# A Combined Cyanine/Carbomer Gel Enhanced Photodynamic Antimicrobial Activity and Wound Healing

**DOI:** 10.3390/nano12132173

**Published:** 2022-06-24

**Authors:** Ming Guan, Guangyu Chu, Jiale Jin, Can Liu, Linxiang Cheng, Yi Guo, Zexing Deng, Yue Wang

**Affiliations:** 1Spine Lab, Department of Orthopedic Surgery, The First Affiliated Hospital, Zhejiang University School of Medicine, Hangzhou 310003, China; guanm@zju.edu.cn (M.G.); 12118297@zju.edu.cn (G.C.); 22018143@zju.edu.cn (J.J.); 12018499@zju.edu.cn (C.L.); q773916513@zju.edu.cn (L.C.); 2Shaanxi Key Laboratory of Brain Disorders, Xi’an Medical University, Xi’an 710021, China; guoyi@xiyi.edu.cn; 3College of Materials Science and Engineering, Xi’an University of Science and Technology, Xi’an 710054, China

**Keywords:** cyanine, carbomer, wound repair, antibacterial property, PDT

## Abstract

As a non-invasive and non-specific therapeutic approach, photodynamic therapy (PDT) has been used to treat antibiotic-resistant bacteria with encouraging efficacy. Inspired by light, the photosensitizers can produce excessive reactive oxygen species (ROS) and, thus, effectively destroy or kill bacteria. Cyanine (Cy), a traditional photosensitizer for PDT, has the advantages of low cytotoxicity and high ROS yield. Yet, the water solubility and photostability for Cy are poor, which substantially limit its antibacterial efficiency and clinical translation. Herein, we combined Cy with carbomer gel (CBMG) to form a photodynamic Cy-CBMG hydrogel. In this system, Cy was evenly dispersed in CBMG, and CBMG significantly improved the water solubility and photostability of Cy via electrostatic interactions. The developed Cy-CBMG gel had less photodegradation under laser irradiation and thus can effectively elevate ROS accumulation in bacteria. The Cy-CBMG compound presented remarkable ROS-induced killing efficacy against methicillin-resistant *Staphylococcus aureus* (93.0%) and extended-spectrum *β*-lactamase-producing *Escherichia coli* (88.7%) in vitro. Moreover, as a potential wound dressing material, the Cy-CBMG hydrogel exhibited excellent biocompatibility and effective antimicrobial ability to promote wound healing in vivo. Overall, this work proposed a practical strategy to synthesize a photosensitizer–excipient compound to enhance the photophysical property and antibacterial efficacy for PDT.

## 1. Introduction

Antimicrobial resistance (AMR) is a serious threat for global health and development [1]. Secondary to ischemic heart disease and stroke, AMR was ranked as the third leading Global Burden of Diseases [2]. Due to the accelerated emergence of bacterial antibiotic resistance, antibiotics are becoming less effective, and infections often become invasive or systemic [3]. In this post-antibiotic era, it is essential to seek novel therapeutics for treating various infections with AMR [4,5]. To this end, great efforts have been made to develop new treatment strategies, such as photodynamic therapy (PDT), photothermal therapy, and novel antimicrobial materials [6,7].

PDT is a photochemistry-based treatment that combines photosensitizers, light, and oxygen. When the photosensitizer is activated by the light of a specific wavelength, local reactive oxygen species (ROS) are elevated to destroy the proteins, nucleic acids, and membranes of the cells, and eventually leads to cell death [8,9,10]. Clinically, PDT is used to treat a variety of diseases, including cancer, acne, psoriasis, and bacterial infections [8,11,12]. In animal and pre-clinic studies, there are numerus studies that applied PDT to treat infected wounds, which are a common occurrence in clinics [9,11]. In fact, bacterial infection in skin wound causes unexpected tissue damage and triggers severe inflammatory reactions, which eventually delays wound repair. Although various photosensitizers have been developed to prevent wound infection, the biocompatibility remains a clinical concern. For example, some photosensitizers, such as silica, gold, and iron oxide nanoparticles, are not biodegradable and may cause harm to lives [13,14].

Cyanine (Cy), a structural analog of indocyanine green authorized by the Food and Drug Administration (FDA), is an efficient photosensitizer for PDT [15,16]. With low cytotoxicity and high ROS yield, Cy is tailored to biological uses spanning from experimental imaging to therapy [17]. Yet, its poor water solubility and photostability limit the clinical translation of Cy. To improve performance, Cy was modified by introducing hydrophilic units or heavy metals, but this resulted in decreased biocompatibility in return [18,19]. A new strategy is needed to modify Cy for potential clinical application.

The stabilization of the drugs–excipients compound can be enhanced by a number of physical interactions, such as electrostatic interactions, hydrogen bonding, and hydrophobic interactions [20]. While Cy carries positive charges, a candidate excipient for Cy is carbomer, which is a negative charge carrier [21]. Protonated carboxylic groups in carbomer can link to the molecular network of polyelectrolyte and thus enhance the compatibility and biostability of insoluble drugs [21]. While carbomer is a common adjuvant and used as a thickener, adhesive and stabilizer in clinics [22,23], in theory, carbomer and Cy can form stable gels through electrostatic interactions.

In the current study, we synthesized a combined Cy–carbomer hydrogel compound (Cy-CBMG) with enhanced performance of the solubility and photostability. As expected, the developed biocompatible Cy-CBMG can generate excessive ROS under laser irradiation to kill multidrug-resistance bacteria, and then accelerate wound healing, suggesting its possible application in wound dress (Figure 1). 

### 1.1. Materials

1,3-Diphenylisobenzofuran (DPBF) and 2,2,6,6-tetramethylpiperidine (TEMP) were purchased from Adamas (Shanghai, China). Carbomer 940 (in powder, 99.0%) and triethanolamina (in liquid, >99.5%) were purchased from Macklin Reagent Co. Ltd. (Shanghai, China), and Cy (CAS 523-42-2; in powder, >98.0%) was purchased from Meryer (Shanghai, China). 2,7-Dichlorofluorescein diacetate (DCFH-DA) dye and CCK-8 kit were purchased form Sigma-Aldrich (St. Louis, MO, USA). A LIVE/DEAD BacLight Bacterial Viability kit was purchased from Thermo Fisher Scientific (Waltham, MA, USA). Extended-spectrum *β*-lactamase-producing *Escherichia coli* (ESBL *E. coli*, ATCC35218) and methicillin-resistant Staphylococcus aureus (MRSA, ATCC43300) were given courtesy of the Lab of Microbiology at the author’s institute.

### 1.2. Preparation of Cy-CBMG

CBMG was prepared according to a previous report [24]. In brief, 0.25, 0.5, 1, and 2 g carbomer was dissolved in 100 mL double-distilled H_2_O (ddH_2_O) and stirred for 30 min at 60 °C in a water bath. Then, 0.25 mL triethanolamine was used to adjust the final pH to 6.5–7.5. Then, Cy in powder form was added in CBMG to prepare 0.03, 0.3, 1, 3 mM Cy-CBMG; the mixture was stirred for 30 min and placed at 37 °C for 24 h. Meanwhile, CBMG was also placed at 37 °C for 24 h. The prepared CBMG and Cy-CBMG were rinsed with ddH_2_O 3 times and stored in sealed glass tube at room temperature.

### 1.3. The Characterization of Cy-CBMG

The micromorphology of CBMG (0.5 wt%) and Cy-CBMG (CBMG 0.5 wt%; Cy 0.3 mM) were characterized by scanning electron microscope (SEM; Regulus8100, Hitachi, Tokyo, Japan) at 10.0 kV accelerating voltage. Briefly, the prepared hydrogels were pre-frozen at −80 °C overnight, then dry-frozen for 48 h. After that, the dry-frozen hydrogels were sputter-coated with gold under vacuum for 10 min before SEM observation.

The swell ratio of CBMG (0.5 wt%) and Cy-CBMG (CBMG 0.5 wt%; Cy 0.3 mM) was measured for 70 h in the PBS. In brief, the freeze-dried CBMG and Cy-CBMG (200 mg, W_0_) were stored in sealed glass tube with 10 mL PBS at 37 °C. At setting time points, the swollen hydrogels were retrieved, blotted dry with a paper towel and weighed (W_S_), then freeze-dried again. The swell ratio was calculated from the following equation:Swell ratio = (W_S_ − W_0_)/W_0_ × 100%

The releasing profile of Cy from Cy-CBMG was measured for 24 h in the PBS. Specifically, 5 g Cy-CBMG hydrogel (0.3 mM) was immersed in 20 mL PBS in glass tube. The tubes were sealed and maintain in incubator at 37 °C. At setting time points (0, 1, 6, 12, and 24 h), 200 μL of supernatant was withdrawn for HPLC-MS/MS, followed by replenished with equivalent fresh PBS.

### 1.4. UV-Vis Absorption Spectroscopy (UV-Vis)

UV-vis absorption spectra were measured with a UV-vis spectrometer (UV-2600, Shimadazu Co. Ltd., Shanghai, China). Then, 100 μL of aqueous solution containing 0.3 mM Cy was transferred to a 1 cm quartz cuvette and the absorption spectra was recorded at the wavelength of 200–800 nm.

For general appearance and solubility assessment of CBMG, Cy in water (3 mM) and Cy in CBMG (3 mM) were photographed, respectively. Then, Cy in water and Cy in CBMG were centrifuged, and the top layer of samples was transferred to a 1 cm quartz cuvette to acquire the UV-Vis measurements (wavelength range: 200–800 nm).

For photostability evaluation, 100 μL Cy in water (0.3 mM) and 100 μL Cy in CBMG (0.3 mM) were transferred to a quartz cuvette, respectively. The samples were irradiated with a 600 nm laser (100 mW/cm^2^). At setting times (0, 3, 6, 9, 12, 15 min), the UV-Vis measurements of the compound samples were acquired at the wavelength of 200–800 nm.

### 1.5. Electron Paramagnetic Resonance (EPR)

The production of ^1^O_2_ (a typical ROS) was detected with an EPR spectrometer (EMXplus-9.5/12, Bruker, Billerica, MA, USA), using TEMP as a spin-trapping probe. Typically, 1 μL TEMP was mixed with 100 μL aqueous solution containing 0.3 mM Cy. The solution was irradiated with a 600 nm laser (100 mW/cm^2^) for 15 min and then immediately transferred to the spectrometer for signal acquisition.

### 1.6. Biocompatibility Assessment

To evaluate the biocompatibility of compound samples, human umbilical vein endothelial cells (HUVECs) were cultured with PBS, carbomer (0.5 wt%), Cy (0.3 mM), Cy-CBMG (0.3 mM) at a concentration of 1 × 10^5^ cells/well in 24-well-plate for 24 h. After washing with PBS for 3 times, cells were harvested and immobilized with 4% paraformaldehyde for 15 min, infiltrated with 0.1% Triton X-1000 for 10 min, and then sealed with 0.3% bovine serum albumin for 30 min. Cells were incubated in Alexa Phalloidin solution (30 μL/mL of PBS-T) for 30 min for actin cytoskeleton staining, and immersed in DAPI (1:1000 *v*/*v*) for 10 min for nucleus staining. After incubation, the cells were washed with PBS 3 times and studied under a confocal laser scanning microscope (Olympus FV3000, Tokyo, Japan).

CCK-8 kits were used to investigate cell viability for samples at different concentrations of. In brief, HUVECs at a concentration of 1 × 10^4^ cells/well was cultured with PBS (control), CBMG (0.25, 0.5, 1, 2 wt%), Cy (0.03, 0.3, 1, 3 mM), Cy-CBMG (0.03, 0.3, 1, 3 mM) for 24 h, respectively. Cells were then washed and stained with 10% CCK-8 solution for 4 h. The relative cell density was measured using a microplate reader at an absorbance value of 450 nm.

### 1.7. Evaluation of Laser-Triggered Antibacterial Activity In Vitro

MRSA (G^+^) and ESBL *E. coli* (G^−^) were employed in experiments. Bacterial cells were cultured in trypticase soy broth (TSB) medium overnight and harvested at the exponential growth phase. The bacteria concentrations were adjusted to 1 × 10^8^ CFUs/mL after centrifuged and re-suspended with fresh PBS.

PBS, Cy (0.03, 0.3, 1, 3 mM), CBMG (0.5 wt%), and Cy-CBMG (0.03, 0.3, 1, 3 mM) were transferred into 1 mL MRSA or ESBL *E. coli* suspensions (1 × 10^6^ CFUs/mL) and then irradiated with laser (600 nm, 100 mW/cm^2^) for 30 min. The control groups did not undergo laser irradiation. The antibacterial activity was assessed using the spread-plate method, ROS analysis, scanning electron microscope, and live/dead staining.

The spread-plate method was used to count the viable bacteria after various treatments. In brief, after being diluted serially, 100 μL bacterial suspension was spread on a sheep blood agar plate and cultured overnight. The number of bacterial colonies was counted for each sample. Bacterial viability was calculated using the following equation: Bacterial viability (%) = E/C × 100%, where C is CFUs in PBS (laser-) group and E represents CFUs in the experiment group.

Using 2,7-dichlorofluorescein diacetate (DCFH-DA) dye, ROS analysis was performed to measure the intracellular ROS levels in bacteria after various treatments. Specifically, the as-prepared MRSA were incubated with 10 μM DCFH-DA at a dark room for 30 min. After rinsing twice, the luminescence intensity was detected by a microplate reader (488/525 nm). Then, the stained bacteria were imaged with a confocal laser scanning microscope.

The morphology changes of MRSA and ESBL *E. coli* were determined with a SEM. As-prepared bacteria were washed twice with PBS and fixed with 2.5% glutaraldehyde at 4 °C for 4 h. After that, the serially dehydration was performed in ethylalcohol–water hybrid solution (30, 50, 70, 85, 90, 95, 100%) for 10 min, respectively. Then, bacteria were freeze-dried and sputter-coated with gold before SEM (Regulus8100, Hitachi, Japan) observation.

A LIVE/DEAD BacLight Bacterial Viability Kit was used to examine bacterial viability after treatments. According to the manufacturer’s protocol, STYO-9 and propidium iodide (PI) dyes were used to stain live and dead cells with green and red luminescence, respectively. In brief, the treated bacteria were harvested and stained with dyes (1:1000 in saline for both) for 20 min in a dark room. The collected bacteria were then rinsed with saline for twice, and then confocal microscope (Olympus FV3000, Tokyo, Japan) observations were performed.

### 1.8. Evaluation of Antibacterial Activity of Cy-CBMG In Vivo

A mouse model of MRSA-infected wound was used to assess antibacterial activity of the synthesized materials. A total of 80 female Balb/c mice (6-week-old, ∼20 g), which were fed in an aseptic environment, were used. Mice were anesthetized by intraperitoneal injection with 3% pentobarbital sodium. The dorsal hair was shaved and sterilized to create a 6 mm × 6 mm full-thickness skin wound using a surgical blade. The MRSA suspension was grown overnight and then collected after rinsing with fresh PBS. The MRSA suspension (1 − 2 × 10^7^ CFUs/cm^2^) was then smeared in the wound.

Two days later, the mice were randomly divided into 8 groups (10 mice/group) based on laser irradiation exposure and treatments. Before treatment, the wound was photographed, and the time was defined as day 0. For 4 groups, mice were given the following treatments at day 0 and day 1: PBS, Cy (0.3 mM), CBMG (0.5 wt%), or Cy-CBMG (0.3 mM), and further received 600 nm laser irradiation (100 mW/cm^2^) for 30 min (once a day for 2 days). Mice in the other 4 groups were treated with the abovementioned medicines but did not undergo laser irradiation. The effectiveness of the various treatments was evaluated at setting times.

On day 3, 5 mice were randomly selected in each group for bacteriological and histological assessments. The mice were sacrificed using overdose anesthesia, and the tissues around the wound were harvested and cut into two halves. One tissue piece was immersed in 2 mL PBS and homogenized for bacteriological evaluation using spread plate method, and another was fixed in 4% glutaraldehyde and embedded for histological study, including HE and Giemsa staining.

On days 4, 8, and 12, the wound was photographed, and the area was measured with ImageJ (V1.8.0, NIH, Bethesda, MD, USA) to assess the healing process. On day 12, all mice were executed using overdose anesthesia, and the wound tissues were collected for Masson trichrome staining. Meanwhile, vital organs (heart, liver, spleen, lung, and kidney) were harvested for HE staining and histological evaluation.

### 1.9. Statistics and Reproducibility

Data were reported as mean ± standard deviation (SD). One-way or two-way analysis of variance (ANOVA) and *t*-tests were used for data analysis, as appropriate. *p* < 0.05 was considered statistically significant.

## 2. Results and Discussion

In this work, we combined carbomer with Cy to form Cy-CBMG hydrogels via electrostatic interactions. Cy-CBMG hydrogels were found to have increased solubility, elevated photostability, and enhanced photo-activated antibacterial activity both in vitro and in vivo, as compared with Cy in water.

### 2.1. Photophysical Characterization of Cy-CBMG Hydrogels

The micromorphology of CBMG and Cy-CBMG was characterized by SEM. As shown on Figure 1a, both hydrogels exhibited a honeycomb-like microstructure, and there was little difference between CBMG and Cy-CBMG. It was suggested that Cy had little effect on the CBMG morphology, and the 3D porous structure would enable cell penetration and ingrowth [25]. Furthermore, considering that excessive exudates would delay wound healing, the hydrogels with desirable swelling capacity to absorb exudates are great candidates for wound care. The swell ratio of CBMG and Cy-CBMG was measured for 70 h. As shown on Figure 1b, a swelling equilibrium is reached after immersed in PBS for 10 h. The swelling ratio of Cy-CBMG (c.a 801%) was slightly lower than the value of CBMG (c.a 865%), which might be attributed to the electrostatic interaction between Cy and CBMG. Cy might provide non-covalent cross-linking through iron pairs to improve the gel network stability, thus resulting in a lower swelling ratio of hydrogels [25]. Moreover, to evaluate the release behavior of Cy in vitro from Cy-CBMG, the accumulative release profile of Cy was measured by HPLC-MS/MS. As shown in Figure 1c, the cumulative release amount of Cy at 1 and 24 h after incubation was 44.0 (c.a 0.1%) and 210.3 pg/mL (c.a 0.5%), respectively. In a word, the release behavior of Cy can be summarized as a relatively sustained release over 24 h. These results indicated that Cy-CBMG was successfully prepared with a 3D porous structure, high swelling ratio and stable interaction between Cy and CBMG, which might be suitable for wound dressing.

The capability of producing ROS was determined for Cy with DPBF absorbance and EPR spectroscopy. DPBF can react with singlet oxygen (^1^O_2_), a typical ROS, and demonstrate a decreased absorption intensity at 410 nm [26]. Therefore, the production of ^1^O_2_ was examined using DPBF as the probe molecule in water. Cy was irradiated by a 600 nm laser (100 mW/cm^2^). As the time of light irradiation increased, the absorption intensity of DPBF around 410 nm gradually decreased (Appendix A). However, no ROS was captured by DPBF in the absence of light or Cy molecules (Appendix A). Further, EPR was used to determine if ^1^O_2_ was produced by Cy under laser irradiation (600 nm), with TEMP as the ^1^O_2_ trapping agent. As shown in Appendix A, the ESR signals for Cy with TEMP under laser irradiation for 15 min clearly displayed a 1:1:1 triplet signal, which was consistent with this for 2,2,6,6-tetramethylpiperidine-N-oxyl (TEMPO). This result confirmed that ^1^O_2_ was produced by Cy.

To investigate the effects of CMBG on the solubility and photostability of Cy, the general view and UV-Vis view of Cy in water and CBMG were photographed and assessed. Results showed that Cy was evenly distributed in CBMG, while there were some precipitations of Cy in water (Figure 1d). The solubility of Cy in water and CBMG was further evaluated by UV-Vis spectroscopy. After centrifugation, 100 μL supernatant from two media (water and CBMG) was dissolved in ethanol and measured by UV-Vis. The UV-Vis absorption for Cy was significantly greater in CBMG than in water (Figure 1e), suggesting that more Cy molecules were dissolved and dispersed in CBMG. The interactions between the positive tertiary amino group in Cy and the carboxyl group in carbomer yielded a high degree of counterion condensation through ion pairs. As such, the drug molecules remain associated in the macromolecular phase, which is with higher viscosity and lower kinetic energy than in the fluid phase [21,27,28].

In addition, the photostability of Cy in water and in CBMG were determined by UV-Vis. Under laser irradiation, Cy in CBMG exhibited excellent photostability with little change in the UV-Vis absorption spectrum (Figure 1f and Appendix A). In contrast, the absorption spectrum of Cy molecules in water gradually decreased, and approximately 9% of the Cy molecules was degraded after 15 min (Figure 1f and Appendix A). Previous studies demonstrated that the amine group in Cy and the carboxylic acid group in carbomer could connect together through electrostatic interactions [21,28]. As the number of effective collision decreased in molecule dispersion, the drugs became more stable upon electrostatic interactions. Based on these results, we concluded that CBMG can promote the solubility and photostability of Cy through electrostatic interactions (Figure 1g).

### 2.2. Biocompatibility of Cy-CBMG Hydrogels

Good biocompatibility is critical for biomaterials from bench to bedside [29]. Biocompatibility of Cy, CBMG, and Cy-CMBG at different concentrations were assessed using CCK-8 and cytoskeleton staining. When the concentration of CBMG was higher than 0.5 wt%, cell viability declined significantly (Figure 2a). Taking the biosafety and loading ability into consideration, 0.5 wt% carbomer was used for further experiments. Unlike carbomer, Cy had little toxicity to cells, even at 10 mM (Figure 2b) which is far beyond the dose for clinical use of PDT [30]. As expected, Cy-CBMG inherited low cytotoxicity from two raw materials and exhibited good biocompatibility. In 10 mM, the cell viability was still greater than 90% (Figure 2b). In cytoskeleton staining, only slight changes in cell morphology were observed after Cy-CBMG (0.3 mM) treatment (Figure 2c), suggesting that this compound was suitable for biological application.

### 2.3. Photo-Activated Antibacterial Performance of Cy-CBMG In Vitro

To test the theory that promoted that the solubility and photostability of Cy in CBMG may enhance its biological application, the antibacterial activity of the Cy-CBMG was investigated. MRSA (Gram^+^) and ESBL *E. coli* (Gram^−^) were used as model bacterial strains. Using the spread-plate method, the cell viability of MRSA and ESBL *E. coli* after various treatments was determined by assessing colony-forming units (CFUs) on each plate. As shown in Figure 3a,b and Appendix A, CBMG, Cy, and Cy-CBMG had low dark cytotoxicity against MRSA and ESBL *E. coli*. However, photo-activated Cy and Cy-CBMG exhibited dose-dependent antibacterial activities for MRSA and ESBL *E. coli*. Taking the biosafety and bactericidal effectiveness of Cy into account, the feeding concentration at 0.3 mM was selected for further evaluation. In this system, photo-activated Cy-CBMG exhibited excellent antibacterial efficiency (93.0%) towards MRSA, which far outweighed Cy dissolved in water (80.7%) and other groups (Figure 3a,b). Similar antibacterial effect was observed toward ESBL *E. coli* (88.7%, Appendix A).

The antibacterial mechanism of photo-activated Cy-CBMG was further investigated. The over-production of ROS was a dominant antibacterial mechanism for PDT [31]. The intracellular ROS was detected using a probe DCFH-DA. After entering the cell, DCFH-DA is deacetylated by cellular esterases into nonfluorescent DCFH. In the presence of ROS, DCFH is further transformed into highly fluorescent 2,7-dichlorofluorescein (DCF) [32,33], which can be detected by a confocal laser scanning microscope and microplate reader under 488/525 nm. More accumulation of ROS in bacterial cells was observed in photo-activated Cy and Cy-CBMG, as compared with other groups (Figure 3c). Strikingly, ROS was upregulated by more than 2-fold in photo-activated Cy and approximately 3-fold in Cy-CBMG under laser irradiation, as compared with other groups (Figure 3d). In a cell, there is a sophisticated system against ROS, including enzymes, such as superoxide dismutase, catalase, glutathione peroxidase [34]. When this system is unable to defend upregulated ROS effectively, the membranes, cytoskeleton, protein and nucleic acids of the cells will be irreversibly damaged [9]. Data suggested that better antibacterial efficiency in photo-activated Cy-CBMG was attributable to the effective yield of ROS, which was in accordance with better photostability in Cy-CBMG after prolonged laser exposure (Figure 1f).

In addition, the morphological changes of MRSA and ESBL *E. coli* were studied under SEM. Figure 4a and Appendix A display the morphology of bacteria with or without laser irradiation. The normal MRSA and ESBL *E.coli* had smooth and intact cell walls. In Cy-CBMG with laser irradiation, collapsed and distorted cell membranes were commonly observed, as was further verified by bacterial live/dead staining. After treatments, live cells were stained by STYO-9 with green fluorescence, while the dead cells, which decreased membrane impenetrability, were labeled with red fluorescence by PI. As shown in Figure 4b, faint green fluorescence and apparent red signals were observed in photo-activated Cy-CBMG, indicated that Cy-CBMG under laser irradiation had produced excessive ROS to damage the integrity of bacterial membrane, which eventually led to cell death. Results demonstrated that Cy-CBMG had outstanding antibacterial efficacy and deserved further study in vivo.

### 2.4. Wound Disinfection and Repair In Vivo

As a potential wound dressing material, the developed Cy-CBMG hydrogel was tested for wound healing efficacy (Figure 5a). Upon the limited penetration of laser irradiation [35,36], a superficial skin infection model of mice was established by implanting MRSA germs on a dorsal wound. While the skin has the capability of self-regeneration, which involves a series of biological processes of hemostasis, inflammation, proliferation, and remodeling, the presence of bacteria can ignite persistent inflammation, which may damage the healing cascade and eventually lead to delayed tissue regeneration and wound healing [37,38]. Hence, wound disinfection is an essential approach to reboot the repairing system of the skin [39].

After various treatments, the CFUs of MRSA within infected skin tissues are presented in Figure 5b. There were numerous bacteria in these groups without laser irradiation. With laser irradiation, the PBS and CBMG groups displayed similar amounts of CFUs to those without laser exposure (Figure 5b). Interestingly, CFUs decreased significantly in Cy treatment with laser exposure, and the bacteria were nearly completely scavenged (>99.5%) in the treatments of Cy-CBMG with laser irradiation (Figure 5c), which was in line with the antibacterial activity in vitro. In addition, Giemsa staining also confirmed that bacteria were eliminated within wound tissues after being treated with Cy-CBMG and laser irradiation (Figure 5d). The results highlight that the photoactive Cy-CBMG hydrogels were able to effectively eradicate MRSA in vivo.

It is reported that persistent inflammation may impair wound healing in infected wound model or diabetes model [40,41]. HE staining was performed to investigate inflammatory cell infiltration in each group. As shown in Figure 5e, a severe inflammatory response was triggered in the wound tissues in all laser-free groups and in PBS and CBMG groups with laser irradiation, as was evidenced by the penetration of numerous white blood cells. Cy with laser irradiation can attenuate the inflammation of wound tissues, and Cy-CBMG with laser irradiation further reduced the inflammation response, as compared to Cy with laser irradiation. In an infected wound, the components of bacteria are highly immunogenic, which can trigger host immune responses [37]. Hence, photo-activated Cy-CBMG can effectively reduce inflammatory responses in the target area. The findings demonstrate that the photo-activated Cy-CBMG had excellent anti-infective ability and thus may facilitate wound healing.

Within 12 days, the wound was photographed at days 0, 4, 8, and 12. Over time, the wound area in mice tended to reduce in all groups. At day 4, the wound healing rate was 74.8% in the Cy-CBMG/laser group, which was significantly higher than in the Cy + laser irradiation group (53.8%). For other groups, the wound healing rate was lower than 30% at day 4 (Figure 6a,b). In addition to effective wound disinfection function by PDT, Cy-CBMG shielded wounds against bacterial re-invasion and maintained a moist microenvironment, which are beneficial for wound healing [42].

While the wound closure rates in Cy-CBMG/laser group and Cy/laser group were nearly 100% and 87.4% in day 12, respectively, poor wound healing rates were observed in other groups (<75%). Collagen formation is a hallmark of wound regeneration and skin reconstruction [43]. As shown in Figure 6c, Masson’s trichrome staining indicated that the regenerated skin tissues in Cy-CBMG/laser group displayed a shorter wound gap (Figure 6d) and denser collagen fiber deposition (Figure 6e), as compared with other groups. The findings suggest that photo-activated Cy-CBMG can significantly accelerate wound healing.

In addition, HE staining of heart, liver, spleen, lung and kidney indicated that no obvious tissue damage was observed in mice in all groups (Appendix A), suggesting that Cy-CBMG is biologically safe. Overall, the results suggest that the photo-activated Cy-CBMG hydrogel is a good candidate material for treating infected skin wound.

## 3. Conclusions

The novel compound Cy-CBMG hydrogel developed in the current study, as formed upon electrostatic interactions between Cy and carbomer, had improved solubility and photostability. The laser triggered Cy-CBMG effectively yielded ROS and efficiently suppressed AMR bacterial infection in vitro, with antibacterial efficiency of 93.0% for MRSA and 88.7% for ESBL *E. coli*. Moreover, as a wound dressing material, the Cy-CBMG hydrogel exhibited excellent biocompatibility and effective antimicrobial capability to accelerate wound healing (wound closure rate was close to 100% after 12-day treatment) in vivo. Results demonstrate that the Cy-CBMG hydrogel is a promising alternative material for antibiotic dressing for infected wounds.

## Data Availability

The data presented in this study are available on request from the corresponding author.

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
