# Peer review of "A Combined Cyanine/Carbomer Gel Enhanced Photodynamic Antimicrobial Activity and Wound Healing"

_nanomaterials, 2022, doi:10.3390/nano12132173_

Round 1

Reviewer 1 Report

In introduction part.

1. Theres is too little introduction for the PDT of antibacterial and wound healing reports. Furthermore, the main purpose of this should be more introduced in the introduction section.

2. The sources of each chemicals should be indicated in the separed Material section (1.1. Materials) and (the source, company, city, country) 

Results section

3. There is no drug release profile from hydrogel. Drug release study should be done.

4.  If the effect of inflammation on the migration of cells are added to support Figure 5, these studies must be more helpful to readers.

Author Response

Reviewer 1#

  1. Theres is too little introduction for the PDT of antibacterial and wound healing reports. Furthermore, the main purpose of this should be more introduced in the introduction section.

Response 1: We express our sincere thanks to the reviewer for his/her professional comment. Indeed, we agree with the reviewer that the relation of PDT of antibacterial and wound healing was introduced too little in introduction section. The relative contents were added in the revised manuscript in Page 2 Line 46-51. To better express the main idea and research purpose of our manuscript, we have modified the introduction in Page 2 Line 69-73.

  1. The sources of each chemicals should be indicated in the separed Material section (1.1. Materials) and (the source, company, city, country)

Response 2: Thank you very much for the professional suggestion. The sources of each chemicals used in this assay have shown in Material section (1.1. Materials).

  1. There is no drug release profile from hydrogel. Drug release study should be done.

Response 3: We appreciate this suggestion and give a further explanation. The cumulative release amount of Cy over 24 h was added in Figure S1.

  1. If the effect of inflammation on the migration of cells are added to support Figure 5, these studies must be more helpful to readers.

Response 4: We appreciate the professional suggestion and give a further explanation. We agree that it is helpful for reader to clearly articulate the effect of inflammation on the migration of cells in this work. Undoubtedly, the migration of cells would be disturbed in the unexpected context of inflammation, such as endothelial progenitor cells, dermal fibroblasts, macrophages, neutrophils and so on1-3. The  prevailing view is that persistent bacterial infection would trigger excessive inflammation, which in turn causes delayed wound healing. Therefore, numerous studies as we do have carried out to combat bacterial infection, which has proved to be effective in wound healing process4, 5. In other words, the acceleration of wound healing is deeply based on reduction of bacterial infection, and alleviation of inflammation is just a subsequent result. In fact, our current study mainly focus on how to improve the water solubility and photostability of cyanine to enhance its antibacterial efficiency and clinical translation. The combined general observation, histological, and bacteriological evidence in our assay is clearly indicated that Cy-CBMG would be promising wound dress material for wound care. Thus, we authors all agree that the effect of inflammation on the migration of cells should be treated as an interesting but independent work for a further study.

Reference

  1. Lee, J. H.; Ji, S. T.;  Kim, J.;  Takaki, S.;  Asahara, T.;  Hong, Y. J.; Kwon, S. M., Specific disruption of Lnk in murine endothelial progenitor cells promotes dermal wound healing via enhanced vasculogenesis, activation of myofibroblasts, and suppression of inflammatory cell recruitment. Stem cell research & therapy 2016, 7 (1), 158.
  2. Beyer, S.; Koch, M.;  Lee, Y. H.;  Jung, F.; Blocki, A., An In Vitro Model of Angiogenesis during Wound Healing Provides Insights into the Complex Role of Cells and Factors in the Inflammatory and Proliferation Phase. International journal of molecular sciences 2018, 19 (10).
  3. Mi, Y.; Zhong, L.;  Lu, S.;  Hu, P.;  Pan, Y.;  Ma, X.;  Yan, B.;  Wei, Z.; Yang, G., Quercetin promotes cutaneous wound healing in mice through Wnt/β-catenin signaling pathway. Journal of ethnopharmacology 2022, 290, 115066.
  4. Maleki, A.; He, J.;  Bochani, S.;  Nosrati, V.;  Shahbazi, M. A.; Guo, B., Multifunctional Photoactive Hydrogels for Wound Healing Acceleration. ACS Nano 2021, 15 (12), 18895-18930.
  5. Chang, M.; Nguyen, T. T., Strategy for Treatment of Infected Diabetic Foot Ulcers. Accounts of chemical research 2021, 54 (5), 1080-1093.

Reviewer 2 Report

The manuscript reports on the use of cyanine dye (Cy) combined with carbomer gel (CBMG) to form a photodynamic Cy-CBMG hydrogel. Synergetically, Cy was evenly dispersed in CBMG, and CBMG significantly improved the water solubility and photostability of Cy via electrostatic interactions. The Cy-CBMG mixture showed remarkable ROS-induced killing efficacy against methicillin-resistant Staphylococcus aureus (93.0%) and Extended-spectrum β-lactamase-producing Escherichia coli (88.7%) in vitro.

The manuscript is interesting in proposing a practical strategy to prepare formulations and wound dressing materials containing photosensitizing dyes to improve the PDT antibacterial efficacy of the Cy dye.

It is necessary to review the English text, for example:

L16, “antibiotic-resistance bacteria” should be changed to antibiotic-resistant bacteria.

L81-82, “was dissolved in 100 mL ddH20 to form various concentrations of hydrogels. Then, triethanolamine was used to neutralize the pH to 6.5-7.5.”

It should be replaced by:

was dissolved in 100 mL of doubly destilled / distilled and deionized ??? H2O to prepare hydrogels of various concentrations. Then, triethanolamine was used to adjust the pH to 6.5-7.5.

Also check the size of some words throughout the manuscript.

Some more important issues:

Is it appropriate to use the concept “laser-inspired” for a laser-driven PDT treatment? Please, modify the entire text of the manuscript (including also the title of the manuscript).

Section 2.1. Photophysical characterization of Cy-CBMG hydrogels.

“As shown in Figure 1c, Cy clearly displayed 1:1:1 triplet EPR signal, indicating that 1O2 was produced by Cy.”

Is really the Cy photosensitizer or, more precisely, the (ROS-modified) TEMP probe, the actual responsible for the triplet EPR signal? Please, make the appropriate corrections and improve the explanation.

Figure S1, please, specify the time interval (minutes).

Please check the names (including the capital letters) and correct the abbreviations and punctuation of several journals included in the References section.

Author Response

Reviewer 2#  

  1. It is necessary to review the English text, for example: L16, “antibiotic-resistance bacteria” should be changed to antibiotic-resistant bacteria. L81-82, “was dissolved in 100 mL ddH20 to form various concentrations of hydrogels. Then, triethanolamine was used to neutralize the pH to 6.5-7.5.” It should be replaced by: was dissolved in 100 mL of doubly destilled / distilled and deionized ??? H2O to prepare hydrogels of various concentrations. Then, triethanolamine was used to adjust the pH to 6.5-7.5. Also check the size of some words throughout the manuscript.

Response 1: We appreciate the professional suggestion and give a further explanation. We have reviewed the English text and the changes are shown in red text in the revised manuscript

  1. Is it appropriate to use the concept “laser-inspired” for a laser-driven PDT treatment? Please, modify the entire text of the manuscript (including also the title of the manuscript).

Response 2: We appreciate the professional suggestion and give a further explanation. The words of “laser-inspired” were all replaced with “photo-activated” in the entire text and specifically it was replaced with “photodynamic” in title, which may more fit the key idea in this manuscript.

  1. Section 2.1. Photophysical characterization of Cy-CBMG hydrogels.

“As shown in Figure 1c, Cy clearly displayed 1:1:1 triplet EPR signal, indicating that 1O2 was produced by Cy.”

Is really the Cy photosensitizer or, more precisely, the (ROS-modified) TEMP probe, the actual responsible for the triplet EPR signal? Please, make the appropriate corrections and improve the explanation.

Response 3: We appreciate the professional suggestion and give a further explanation. TEMPO, the ROS-modified TEMP probe, is the actual responsible for the triplet EPR signal. Herein, we modified this section and give better explanation with “As shown in Figure 1c, the ESR signals for Cy with TEMP under laser irradiation for 15 min clearly displaye d 1:1:1 triplet signal, which was consistent with this for 2,2,6,6-tetramethylpiperidine-N-oxyl (TEMPO). This result confirmed that 1O2 was produced by Cy.”

  1. Figure S1, please, specify the time interval (minutes).

Response 4: We appreciate the professional suggestion and give a further explanation. The specific time interval was added in Figure S2.

  1. Please check the names (including the capital letters) and correct the abbreviations and punctuation of several journals included in the References section.

Response 5: We appreciate the professional suggestion and give a further explanation. All references cited in this manuscript were revised.

Reviewer 3 Report

This manuscript describes the preparation of Cyanine/carbomer hydrogel as a potent antimicrobial system for wound healing applications. The manuscript introduction, materials and preparation method sections are lack in details. The final application should be extensively described in the introduction. Authors should also describe what kind of PDTis  usually use with Cy, what type of diseases are treated with this treatment, if the combination Cy-CBMG can be used for the same purpose and the same dose, etc. Besides, they should describe the innovation part of this work.

Does the intense blue color affect to the final application of this hydrogel?

First image is named as scheme 523, why? Is it a copy from another manuscript or book? In the image foot description it is described the material section, why? It should be described along the manuscript text in a material section.

Preparation of the Cy-CBMG hydrogel is not well described: How long do you need to obtain the hydrogel? Which temperature was the hydrogel preparation carried out? How do authors eliminate the residual reactants after gel preparation? Did they use any kind of stirring during the preparation method? What Cy concentrations did they test? Did authors add the Cy in powder or in solution to the CBMG solution? How did authors corroborate that hydrogel was formed properly? They did not conduct any characterization of the hydrogel after its preparation, such as FTIR, SEM, swelling test, etc.

Characterization by NMR is not completely described, for example how many scans did they use? Resolution of the instrument? 

UV-vis characterization is confused, for example firstly authors refer to prepare the CBMG solution at 3mM, but after they mention 0.3mM. What is the absorption peak that authors used to analyze the results? Is the Cy-CBMG soluble in water after gelling?

The biocompatibility assessment section describes equipment such as Freeze-drying and SEM but they have not been described in the preparation methods. What are the experimental conditions used for these procedures and the equipment characteristics?

The text font is not uniform along the manuscript. For example, lines 143, 144, 151, 155, 231, 253 show bigger text font than the rest of the manuscript. 

Legend in figure 1b is wrong. Please correct the Cy+DBPG+laser system by Cy+DPBF+laser.

Figure 1e shows the UV-vis spectra of Cy dissolved in water and in CBMG. Which blank solution did authors use for the measurement of the baseline for Cy-CBMG system?

Why did authors conduct the biocompatibility study in different concentration units for carbomer and for Cy and Cy-CBMG? Carbomer graph is shown in wt% while Cy and Cy_CMBG is shown in mM units. It is convenient to show all of them at the same concentration units.

Author Response

Reviewer 3#  

  1. The manuscript introduction, materials and preparation method sections are lack in details. The final application should be extensively described in the introduction. Authors should also describe what kind of PDT is  usually use with Cy, what type of diseases are treated with this treatment, if the combination Cy-CBMG can be used for the same purpose and the same dose, etc. Besides, they should describe the innovation part of this work.

Response 1: We appreciate the professional suggestion and give a further explanation. More detail information was added in this manuscript introduction, materials and preparation.

  1. Does the intense blue color affect to the final application of this hydrogel?

Response 2: We appreciate the professional suggestion and give a further explanation. As shown on Figure 1d, Cy-CBMG at concentration of 3 mM was exhibited with such intense blue color. Actually, as shown on Figure R, we used 0.3 mM Cy-CBMG with lighter color for our final application in vitro and in vivo.

Figure R. A general view of 0.3 mM Cy-CBMG.

  1. First image is named as scheme 523, why? Is it a copy from another manuscript or book? In the image foot description it is described the material section, why? It should be described along the manuscript text in a material section.

Response 3: We appreciate the professional suggestion and give a further explanation. We note that our scheme graph which should be defined as “Scheme I. Photo-activated Cy-CBMG generates excessive ROS to combat MASR infection and promote wound repair”, but likely due to a typesetting error, was wrong presented as scheme 523 and it was revised on manuscript. Otherwise, the materials section was separated and shown as “1.1 materials”

  1. Preparation of the Cy-CBMG hydrogel is not well described: How long do you need to obtain the hydrogel? Which temperature was the hydrogel preparation carried out? How do authors eliminate the residual reactants after gel preparation? Did they use any kind of stirring during the preparation method? What Cy concentrations did they test? Did authors add the Cy in powder or in solution to the CBMG solution? How did authors corroborate that hydrogel was formed properly? They did not conduct any characterization of the hydrogel after its preparation, such as FTIR, SEM, swelling test, etc.

Response 4: We appreciate the professional suggestion and give a further explanation. The more details about preparation of Cy-CBMG were added in the methods section, and SEM and swelling ratio of CBMG and Cy-CBMG were characterized, and the preparation and results were added in this manuscript. The micromorphology of CBMG and Cy-CBMG were characterized by SEM. As shown on Figure 1a, both hydrogels were exhibited honeycomb-like microstructure, and there was little difference between CBMG and Cy-CBMG. It was suggested that Cy had little effect on CBMG morphology and the 3D porous structure would enable cells penetration and ingrowth1. Furthermore, considering excessive exudates would delay wound healing, the hydrogels with desirable swelling capacity to absorb exudates are great candidates for wound care. The swell ratio of CBMG and Cy-CBMG were measured for 70 h. As shown on Figure 1b, a swelling equilibrium is reached after immersed in PBS for 10 h. The swelling ratio of Cy-CBMG (c.a 865%) was slightly lower than the value of CBMG (c.a 801%), which might be attributed to the electrostatic interaction between Cy and CBMG. Cy might provide non-covalent cross-linking through iron pairs to improve gel network stability, thus resulting in lower swelling ratio of hydrogels1. Moreover, to evaluation the release behavior of Cy in vitro from Cy-CBMG, the accumulative release profile of Cy was measured by HPLC-MS/MS. As shown in Figure 1c, the cumulative release amount of Cy at 1 and 24 h after incubation was 44.0 and 210.3 pg/mL, respectively. In a word, the release behavior of Cy can be summarized as a relatively sustained release over 24 h. These results indicated that Cy-CBMG was successfully prepared with 3D porous structure, high swelling ratio and stable interaction between Cy and CBMG, which might suit for wound dress.

Figure 1. The characteristic of Cy-CBMG and the effects of CBMG on the solubility and photostability of Cy. a. The representative SEM images of CBMG and Cy-CBMG. b. The swell ratio of CBMG and Cy-CBMG at setting time points. c. The releasing behavior of Cy from Cy-CBMG in vitro. d. Images of Cy in water and CBMG. e. The absorption of Cy in CBMG and water dissolved in EtOH. f. Normalized absorbance of Cy irradiated by 600 nm laser in different media. g. Schematic electrostatic interactions between carboxyl group of CBMG and tertiary amino group of Cy.

  1. Characterization by NMR is not completely described, for example how many scans did they use? Resolution of the instrument? 

Response 5: We appreciate the professional suggestion and give a further explanation. Cy (CAS 523-42-2) was purchased from Adamas (Shanghai, China), then we characterized it by NMR for sure. We authors consider that it is no need to present this data in our manuscript, so we delete this section.

  1. UV-vis characterization is confused, for example firstly authors refer to prepare the CBMG solution at 3mM, but after they mention 0.3mM. What is the absorption peak that authors used to analyze the results? Is the Cy-CBMG soluble in water after gelling?

Response 6: We appreciate the professional suggestion and give a further explanation. Due to our careless, the actual concentration we applied was 0.3 mM, which was wrong presented in our manuscript. We had corrected it in our methods section. Cy-CBMG is non-soluble in water after gelling.

  1. The biocompatibility assessment section describes equipment such as Freeze-drying and SEM but they have not been described in the preparation methods. What are the experimental conditions used for these procedures and the equipment characteristics?

Response 7: We appreciate the professional suggestion and give a further explanation. The preparation and the equipment characteristic were added in related section.

  1. The text font is not uniform along the manuscript. For example, lines 143, 144, 151, 155, 231, 253 show bigger text font than the rest of the manuscript. 

Response 8: We appreciate the professional suggestion and give a further explanation. The text font along the manuscript was carefully revised.

  1. Legend in figure 1b is wrong. Please correct the Cy+DBPG+laser system by Cy+DPBF+laser.

Response 9: We appreciate the professional suggestion and give a further explanation. We have corrected the Cy+DBPG+laser system by Cy+DPBF+laser in Figure S1b.

Figure S1. The generation of singlet oxygen (1O2) by photo-activated Cy. a. Time-dependent UV-Vis absorption spectra of DPBF with Cy under laser irradiation. b. Normalized absorbance of DPBF in different conditions. c. EPR spectra of Cy in the presence of TEMP in different conditions.

  1. Figure 1e shows the UV-vis spectra of Cy dissolved in water and in CBMG. Which blank solution did authors use for the measurement of the baseline for Cy-CBMG system?

Response 10: We appreciate the professional suggestion and give a further explanation. EtOH was set as the baseline for Cy-CBMG system and the details were added at methods section.

  1. Why did authors conduct the biocompatibility study in different concentration units for carbomer and for Cy and Cy-CBMG? Carbomer graph is shown in wt% while Cy and Cy_CMBG is shown in mM units. It is convenient to show all of them at the same concentration units.

Response 11: We appreciate the professional suggestion and give a further explanation. Firstly, we perform the different concentrations of carbomer for biocompatibility in order to find the suitable concentration to form a hydrogel, which might be tailor for wound dress. Taking the biosafety and loading ability into consideration, 0.5 wt% carbomer was used for further experiments. Then, the cytotoxic evaluation of Cy and Cy-CBMG with various concentration was performed. Herein, these are two separate experiments, so we showed them with two measurement units and in different summary graphs.

Reference

  1. Song J, Zhang C, Kong S, Liu F, Hu W, Su F, Li S: Novel chitosan based metal-organic polyhedrons/enzyme hybrid hydrogel with antibacterial activity to promote wound healing. Carbohydr Polym 2022, 291:119522.

Round 2

Reviewer 3 Report

Authors have addressed most of the previous comments, however minor revisions are required before accepting for publication:

In the material section, please specify the purity grade of each reactant and the use conditions. 

In the preparation section of Cy-CBMG, what is the concentration of Triethanolamina used for pH adjustment? What is the Cy amount added?

In the discussion section, the swelling ratio of Cy-CBMG and CBMG values are swapped, compared to the graph (figure 1b) and text explanation. 

What is the percentage of cumulative drug released? Please complement the figure 1c with this data.

The statement at line 238 (DPBF can react with singlet oxygen, a typical ROS, and demonstrate a decreased absorption intensity at 410 nm), required some references.

Author Response

Reviewer 3# - Round 2

Authors have addressed most of the previous comments, however minor revisions are required before accepting for publication:

First of all, We are very grateful to the reviewers for their valuable comments, which have greatly improved this manuscript. Again, we produced a point-by-point response to all comments and revised the manuscript carefully.

  • In the material section, please specify the purity grade of each reactant and the use conditions. 

Response 1: We appreciate the professional suggestion and give a further explanation. More detail information about the purity grade of each reactant and the use conditions was added in material section.

  • In the preparation section of Cy-CBMG, what is the concentration of Triethanolamina used for pH adjustment? What is the Cy amount added?

Response 2: We appreciate the professional suggestion and give a further explanation. Triethanolamina (in liquid, > 99.5%) was purchased from Macklin Reagent Co. Ltd (Shanghai, China). Then, 0.25 mL triethanolamine was used to adjust the final pH to 6.5 - 7.5. Then Cy in powder was added in CBMG to prepare 0.03, 0.3, 1, 3 mM Cy-CBMG, and the mixture was stirred for 30min, and placed at 37 °C for 24 h.

  • In the discussion section, the swelling ratio of Cy-CBMG and CBMG values are swapped, compared to the graph (figure 1b) and text explanation. 

Response 3: We appreciate the professional suggestion and give a further explanation. Due to our negligence, the swelling ratio of Cy-CBMG and CBMG values were swapped on Figure 1b. Herein, we corrected it in our results and discussion section and other data presented in this manuscript were fully reviewed.

  • What is the percentage of cumulative drug released? Please complement the figure 1c with this data.

Response 4: We appreciate the professional suggestion and give a further explanation. The percentage of cumulative drug released was complemented. As shown in Figure 1c, the cumulative release amount of Cy at 1 and 24 h after incubation was 44.0 (c.a 0.1%) and 210.3 pg/mL (c.a 0.5%), respectively.”

  • The statement at line 238 (DPBF can react with singlet oxygen, a typical ROS, and demonstrate a decreased absorption intensity at 410 nm), required some references.

Response 5: We appreciate the professional suggestion and give a further explanation. The relative references were added in manuscript.
